# A New General Fatigue Limit Diagram and Its Application of Predicting Die Fatigue Life during Cold Forging

**DOI:** 10.3390/ma15072351

**Published:** 2022-03-22

**Authors:** Man-Soo Joun, Su-Min Ji, Wan-Jin Chung, Gue-Serb Cho, Kwang-Hee Lee

**Affiliations:** 1Engineering Research Institute, School of Mechanical and Aerospace Engineering, Gyeongsang National University, Jinju 52828, Gyeongsangnam-do, Korea; 2Graduate School of Mechanical and Aerospace Engineering, Gyeongsang National University, Jinju 52828, Gyeongsangnam-do, Korea; opzx951@gnu.ac.kr; 3Department of Mechanical System Design Engineering, Seoul National University of Science and Technology, Seoul 01811, Gyeonggi-do, Korea; wjchung@snut.ac.kr; 4Advanced Process and Materials Group, Korea Institute of Industrial Technology, Incheon 21999, Gyeonggi-do, Korea; gscho@kitech.re.kr; 5Pungkang Co., Ltd., Hwaseong 18572, Gyeonggi-do, Korea; kwangheelee@pungkang.co.kr

**Keywords:** cold forging die, general fatigue limit diagram, high cycle fatigue fracture, die life prediction, fatigue functional exponent

## Abstract

Traditional fatigue fracture theory and practice focus principally on structural design. It is thus too conservative and inappropriate when used to predict the high-cycle fatigue life of dies used for metal forming, especially cold forging. We propose a novel mean stress correction model and diagram to predict the high-cycle fatigue lives of cold forging dies, which focuses on the upper part of the equivalent fatigue strength curve. Considering the features of die materials characterized by high yield strength and low ductility, a straight line is assumed for the tensile yield line. To the contrary, a general curve is used to represent the fatigue strength. They are interpolated, based on the distance ratio, when finding an appropriate equivalent fatigue strength curve at the mean stress and stress amplitude between the line and curve. The approach is applied to a well-defined literature example to verify its validity and shed light on the characteristics of die fatigue life. The approach is also applied to practical forging and useful qualitative results are obtained.

## 1. Introduction

Low-cycle fatigue (LCF) in metal forming is deeply related to die life during hot metal forming [1]. To the contrary, the high-cycle fatigue (HCF) life of die parts is of great importance in cold forging [2], especially during automatic multi-stage cold forging. Die fatigue life is becoming a major issue because complicated die structural analysis has become increasingly possible owing to recent advances in metal forming simulation technologies [3,4,5].

Notably, most studies on HCF fractures have focused on structural design. They are thus inclined to conservatively predict fatigue life. For example, the Goodman–Haigh diagram [6,7], which has been widely used by structural engineers, is more conservative than the Gerber diagram. It is thus inappropriate for predicting the HCF life of a cold forging die that will inevitably fracture. Die design in automatic multi-stage cold forging is characterized by die inserts and shrink rings [8]. The die inserts, mostly made up of WC-Co cemented carbide or hard metals [9,10,11,12], are exposed to the HCF fracture.

Few researchers have studied die fatigue fracture during cold metal forming. Knoerr et al. [13] employed a strain-based approach and damage analysis concept to estimate die fatigue life using an elastoplastic finite element method to model the maximum strain amplitude. Sonsöz and Tekkaya [14] studied die fatigue during cold forging both numerically and experimentally, and simulated fatigue crack growth using a finite element method and the Paris/Erdogan fatigue law. Pedersen [15] studied the effect of die prestressing on fatigue fracture using an elastoplastic finite element method and a model of continuum damage mechanics. Falk et al. [16] sought to identify the major factors affecting die HCF life via finite element prediction and found a close association between the tool steel used and the stressed volume in the four-point bending test. Okamoto et al. [12] researched WC-Co, particularly in terms of ductility and brittleness, using cylinder compression and three-point bending and transverse rupture tests. Choi and Kim [17] used Goodman and Gerber diagrams to predict die fatigue life during the hot forging of an AISI1045 socket for a ball joint assembly used in a civilian vehicle.

Fu et al. [18] systematically predicted die fatigue life via both stress-life and strain-life fatigue analysis methods based on the maximum principal stress and strain, respectively. Lee et al. [19] studied the effect of shrink-fitting on die stress in terms of HCF life. Li et al. [20] conducted three-point bending fatigue tests of WC-Co cemented carbides with single edge notches under static and cyclic (20 Hz) loads at room temperature. Wang et al. [21] used an experimental “master fatigue limit diagram” of the die insert material SKD11 [22] to calculate an optimized shrink-fitting ratio for a two-layer compound die in backward extrusion. Tanrikulu and Karakuzu [2] performed experimental and numerical studies to predict quantitatively the fatigue life of a WC-Co20% cold forging die using an improved Goodman–Haigh diagram and the Basquin law [23].

Simple tension and/or compression tests, combined with three- or four-point bending tests using various combinations of mean stress and stress amplitude, have traditionally been conducted. However, the stress conditions under a fully reversed stress (when the stress ratio *R* = −1) are optimal for testing, particularly for metal forming. However, many tests are conducted under conditions where stresses equivalent to the normal stress condition (*R* = −1) are required. Several schemes and traditional MSCDs or fatigue limit diagrams are used to define the equivalent stresses, including the Goodman–Haigh, Gerber, Soderberg, Morrow, Walker, Dietman, Mason–Halford, and Smith–Watson–Topper diagrams, and variants thereof [24,25,26,27]. The Goodman–Haigh diagram is most often used to estimate the effect of mean stress on fatigue strength. However, Niesłony and Böhm [28] developed a reasonably accurate two-S-N curve approach, recently applied by Böhm et al. [29]. Mortazavian and Fatemi [30] studied the effects of mean stress on fatigue behavior of short fiber reinforced thermoplastic composites. Zhu et al. [27] developed a mean stress-corrected strain energy model and performed a variety of experiments. Ince [31] presented a fatigue damage model based on the distortional strain energies of both positive and negative mean stresses. The model was experimentally acceptable. Liu et al. [32] employed the cyclic strain energy density concept to reflect the effect of mean stress on two characteristic fatigue thresholds, i.e., the fatigue limit and the fatigue crack propagation threshold.

The mean stress effect, and modeling thereof, remain of major interest. Most researchers used Goodman–Haigh diagrams, which are conservative (to promote the design of safer structures) and focus on negative hydrostatic pressure in most cases. The die stress during cold metal forming is unlike that in a structural situation, because the die is designed to fracture after a defined number of production cycles and the principal cause of material plastic deformation is compressive stress. The experimental literature [7] shows that the Gerber diagram of fatigue strength is more realistic than the Goodman–Haigh diagram. The latter, however, has been widely utilized to design dies for metal forming, and for structural design. Moreover, the Gerber diagram has a distinct limitation in that it may overestimate die fatigue life when the stress state (for example, at *R* = 0) is near the yield line of the MSCD or fatigue limit diagram.

There are several well-known stress measures [33] for determining one-dimensional stress, for multiaxial stress. The stress is referred to herein as ‘basis stress’ to distinguish it from the stress of continuum mechanics. The simplest basis stress is the absolute maximum principal stress, but its use is inappropriate for predicting the fatigue life of severely prestressed dies. The signed von Mises and Tresca basis stresses assume the magnitude of the effective and maximum shear stresses, respectively, and use the sign of the maximum principal stress. The Sines method [34] considers both the effective stress and hydrostatic pressure when evaluating fatigue. Tanrikulu and Karakuzu [2] considered the tangential stress on the die surface as the basis stress, in accordance with Knoerr et al. [13].

In this paper, we propose a general MSCD or fatigue limit diagram characterized by one or two parameters, i.e., the exponents of the mean stress per unit tensile strength, which not only combine flexibly the Gerber and Goodman–Haigh diagrams, but also modify them. We also extend the diagram to the negative mean stress range using an additional parameter. We use the general MSCD to predict the HCF life of the dies of a nut cold forging process.

## 2. Characteristics of Cyclic Stresses on a Prestressed Die

Variation in die stress during cold forging by an automatic multi-stage cold forging machine was examined for a typical industrial process. Some of the stages for manufacturing an automotive nut are shown in Figure 1. The third stage is discussed here. The carbon steel S25C nut cold forging process shown in Figure 1 is representative of the application case of highly prestressed die parts considering their HCF features. Notably, a press-fitting method is usually employed for this process. The stresses of the die parts marked by S, U, and W in Figure 1 are utilized to reveal the characteristics of die stresses.

Figure 2 shows the flow curve for S25C, obtained via tensile testing and flow curve acquisition at high strain [35]. The solid lines were fitted, and the dotted lines were extrapolated. Notably, flow stresses (yield stresses at the specific strains) up to strains of 1.3 could be obtained for S25C. The frictional stresses were calculated using Coulomb’s law of friction, and the frictional coefficient μ [36] was assumed to be as follows:(1)$$\mu =0.02\left(1+{\bar{\varepsilon }}_{s}/3\right)\;\mathrm{for}\;S25C$$
where ${\bar{\varepsilon }}_{s}$ is the effective strain of the material at the contact interface.

The implicit elastoplastic finite element method with tetrahedral MINI-elements [4,8,37] was used to simulate the process. We assumed that the material obeyed von Mises yield criterion while the die parts were assumed rigid in simulating the forging process. The stresses exerting on the material-die interfaces were calculated utilizing the tractions predicted by the forging simulation with die parts deformation constrained. All die parts were assumed to be elastic during die structural analysis because the materials of the die inserts, i.e., cemented carbides are so hard. The punch velocities were all fixed at unity because the behavior of S25C was assumed to be rate-independent. The materials used to create die parts are shown in Figure 1, colored with yellow and red for WC-Co20% cemented carbide and with the other for tool steel SHK51. Their properties are listed in Table 1. It was assumed that symmetry was maintained during forging. Thus, only some of the material required analysis. Figure 3 shows a typical finite element mesh system of the material and die parts at an instant. The material was discretized into 190,000 tetrahedrons on average and 90,000, 80,000, and 83,000 tetrahedrons were employed for the finite element analysis model for die parts W, S, and U, respectively.

Figure 4 and Figure 5 show the predicted effective strain and effective stresses on the die parts at the final stroke of the stages defined in Figure 1, respectively. The die inserts were prestressed by the shrink ring and thus their initial mean normal stresses were negative. The mean normal stresses of die inserts at the final strokes usually become greater than those at the initial mean normal stresses. They may become positive at the weak points in terms of the HCF fracture. However, the positive mean normal stresses were initially exerted in the shrink ring and their signs were not changed during forging. Therefore, the effective stresses of the shrink ring in Figure 5 look as great as the die inserts even though the shrink rings are much safer than the die inserts in terms of HCF fracture.

We now discuss the characteristics of the basis stress during cold forging with prestressed and non-prestressed dies. We considered the signed effective stress in the analysis [17] just for convenience. Figure 6 shows the paths of the mean stresses and stress amplitudes of selected finite elements of the prestressed die insert (termed S) shown in Figure 5, indicating that the stresses start from the axis of mean stress at the left and move toward the right upper corner at an angle of almost 45°. Some lines deviate slightly from the 45° slope because the mean stress falls below its initial value during forging. In other words, the basis stresses associated with 45° slopes always increase during forging, which mainly relies on compressive stress. The 45° slope is attributable principally to the increase in circumferential stress during forging.

When the prestresses are very high, as for the nut, the initial mean stress begins further from the origin than in the more common case, as can be seen from the comparison in Figure 6. Note that a line connects the initial and maximum stress states experienced by a finite element during forging simulation and that the numbers of the selected elements are multiple of hundred. The lines can thus represent qualitatively the characteristics of the die stresses in metal forming for HCF life prediction. The stresses do not reach the first quadrant. When the die part is simple, the initial range of mean stress is narrower than that of a complicated shape, such as the die part S. This is because it is very difficult to impart uniform compressive stress to internally stepped dies by applying a radial force to the outer surface. Notably, the finite elements on which higher initial mean stresses were exerted experienced higher stress amplitudes, implying that the die material in the strongly prestressed region (i.e., near the internal surface of the die insert) experienced higher compressive stresses.

However, the shrink ring experienced the opposite effects, as can be seen in Figure 7. The initial mean stresses were all positive and the maximum stress amplitudes were about 85% as large as those of the die inserts. These conditions appear to greatly increase the risk of HCF fracture. However, this situation is in fact often safer than that of the die insert, because the stress paths all move along the −45° slope (as can be seen in Figure 7), which lies parallel to the line denoting the tensile yield line. This explains why shrink ring fracture is much less common than die insert fracture, even though the ring stress state lies in the narrower (i.e., more dangerous) region of fatigue fracture. Note that the Goodman–Haigh diagram, and variants thereof, may underestimate the HCF life of shrink rings in particular, because they estimate fatigue life more conservatively on the right side of the MSCD. The shrink rings are characterized by larger effective stresses on their inner interfaces. The forming loads also exert a marked influence on the same regions, implying that the stress amplitudes of the stressed region are greater than those of the outer diametrical surfaces (Figure 7). This further reduces the accuracy of Goodman–Haigh diagram and variants thereof.

On the contrary, the stress paths of the punch which were not prestressed were in the opposite direction of those of the prestressed cases of die inserts (Figure 8). The paths begin at the origin of the (mean stress, stress amplitude)-coordinate system and run toward the second quadrant. The final destinations of mean stress and stress amplitude are mainly located on a line inclined counterclockwise (by 135°) with respect to the axis of mean stress in the second quadrant, because the punch and ejector principally experience compressive stresses. However, a few of the final destinations deviated from the line (Figure 8), associated with finite elements of which basis stresses changed sign. Such elements are usually located near die corners, i.e., where stress is concentrated.

## 3. Generalized Gerber-Goodman Diagram

For high accuracy, the fatigue strength of a material should be obtained by the maximum value of fully reversed stress (*R* = −1) that its standard specimen can withstand for a specified number of cycles (normally 1000 cycles) without any fatigue fracture. However, actual engineering stresses of die in metal forming are multiaxial and the mean stress is not usually zero. In practical terms, an equivalent stress corresponding to the multiaxial stress is useful for predicting the engineering fatigue life from the S-N curve.

In this study, $\sigma_{a}$, $\sigma_{m}$, $\sigma_{FS}$, $\sigma_{YT}$, $\sigma_{YC}$, $\sigma_{F},$ and $\sigma_{TS}$ are stress amplitude, mean stress, fatigue strength, tensile yield strength, compressive yield strength, true stress at the fracture point, and tensile strength, respectively.

Figure 9 [27] shows a typical dimensionless plot of fatigue strength for a steel and the effect of the mean stress and stress amplitude on their equivalent fatigue strength $\sigma_{FS}$. Note that this figure was used for generality considering a wide range of die materials including WC-Co cemented carbide and die steels even though it is conceptual and may be exaggerated. The distribution of fatigue strength points (with non-zero mean stresses) can be fitted using a simple function, called equivalent fatigue strength curve, because they change in a statistically monotonic fashion. Note that the structural member design focuses on the lower side of the equivalent fatigue strength curve. However, in die parts design, the stress combinations of $\sigma_{m}$ and $\sigma_{a}$ are preferentially placed on its upper side. The equivalent fatigue strength curve in the structural engineering is thus optimal when all experimental failure points lie on its upper side. To the contrary, the curve in metal forming is optimal if it passes through the points to minimize the least-squares error. Therefore, the focus of structural and metal forming engineers differs fundamentally. Note also that the right side of the MSCD is more important in structural engineering, while the left side is much more important in metal forming, as emphasized above.

Several fatigue functions meet the fatigue fracture criterion. In Figure 9, the straight line is the Goodman–Haigh line and the curve is the Gerber parabola, formulated, respectively, as follows:(2)$$\frac{\sigma_{a}}{\sigma_{FS}}+\frac{\sigma_{m}}{\sigma_{TS}}=1$$
(3)$$\frac{\sigma_{a}}{\sigma_{FS}}+{\left(\frac{\sigma_{m}}{\sigma_{TS}}\right)}^{2}=1$$
which can be generalized as follows:(4)$$\frac{\sigma_{a}}{\sigma_{FS}}+{\left(\frac{\sigma_{m}}{\sigma_{TS}}\right)}^{\zeta }=1$$
where ζ is the correction factor exponent to the mean stress per unit tensile strength. ζ = 1 and ζ = 2 describe the Goodman–Haigh and Gerber diagrams, respectively. This expression was employed by Mortazavian and Fatemi [30]. However, they neglected the correction factor exponent in their study because the similarity of Equation (4) with Equation (2) was experimentally verified for the short glass fiber polymer composites. It is known that as the hardness of the material increases the correction factor decreases.

The Goodman–Haigh line is more conservative and thus more popular among structural engineers, who must focus mainly on safety in terms of fatigue fracture. However, the Goodman–Haigh diagram exhibits non-negligible differences from the experimental data shown in Figure 9. This is not the best diagram for predicting the HCF life of the prestressed dies used in metal forming because the dies are allowed to fracture after the target lives, unlike structural members. Despite the importance of predicting die life, the technology for doing so was inadequate until recently, because it was difficult to analyze die mechanics with the requisite accuracy. Thus, the modified Goodman–Haigh line is now more popular among metal forming engineers [2,17].

The results of experiments on aluminum alloys and steels are shown in Figure 10 [6,33]. Assuming that the tensile strength is 1.5 to 2.0 times the yield strength [35], the equivalent fatigue strength curves (dashed lines in Figure 10) cannot be described by straight lines connecting the fatigue strength points of the fully reversed stresses (*R* = −1) to the tensile strengths on the axis of mean stress. Figure 10 also shows that the mean stress exerted by compression increases the equivalent fatigue strength. In most studies on structural and die mechanics [2,17], the researchers drew horizontal lines from the fatigue strength points on the $\sigma_{a}$-axis to extend the functions of the equivalent fatigue strength curves to the negative mean stress side. Few researchers neglected that side because it was considered not important, or because the horizontal line promoted a safer structural design that prioritized fatigue strength. Notably, the experimental data of Figure 10 (approximated by the dot lines) are quite different from those yielded by the presumed Goodman–Haigh lines, which begin along with the assumed tensile strengths on the $\sigma_{m}$-axis, that is, 1.5-fold and 2.0-fold the yield strength. As the yield and tensile strengths of hard materials, such as those of the WC family used in cold forging, are near-identical, the equivalent fatigue strength curve should be more flexible.

In comparison, the Gerber parabola provides a relatively good fit with the experimental data obtained when the mean stress is positive (Figure 9). Most researchers (although not Choi and Kim [17]) used the Goodman–Haigh diagram because it is conservative and linear. Note that the life target of most dies used for metal forming is less than $10^{6}$ cycles at the fatigue strength. Therefore, the accurate prediction of die fatigue life is of great importance for safety, which is the prime consideration in structural engineering.

The curved diagram seems to better describe the equivalent fatigue strength when the mean stress is positive and the material is not so hard (Figure 9). Apparently, the Gerber diagram does not meet the straight line connecting the $\sigma_{YT}$ point on the $\sigma_{a}$-axis and $\sigma_{TS}$ point on the $\sigma_{m}$-axis. In case of the die materials, neither Goodman nor Gerber diagram can thus be representative.

A similar curve can be used to calculate the equivalent stress ($\sigma_{eq}$ defined in Figure 11) near the equivalent fatigue strength curve. However, this curve cannot be used to evaluate extreme points near the tensile and compressive yield lines defined, respectively, as follows:(5)$$\sigma_{a}=-\sigma_{m}+\sigma_{YT}$$
(6)$$\sigma_{a}=\sigma_{m}+\sigma_{YC}$$
which prevent a stress state from violating the tensile and compressive uniaxial yield strengths, respectively.

Therefore, we combined the Gerber and Goodman–Haigh diagrams to define the $\sigma_{a}$-$\sigma_{m}$ curve at the point C in Figure 11. The equivalent fatigue strength curve is thus described by the modified Gerber diagram. The line denoting the tensile yield line or the line connecting $\sigma_{YT}$ on the $\sigma_{a}$-axis and $\sigma_{TS}$ on the $\sigma_{m}$-axis (two lines are almost the same in case of brittle die materials like WC-Co) is the same as that in the Goodman–Haigh diagram. The fatigue functional exponent is interpolated using the fractional lengths *r* and 1 − *r*, as defined in Figure 11, to calculate the target point ($\sigma_{m},\sigma_{a}$) as follows:(7)$$r=\frac{\bar{AC}}{\bar{AB}}$$

This target point C in Figure 11 changes. To ensure generality, the fatigue functional exponents at the two extreme points A and B of Figure 11 are denoted as $\zeta_{1}$ and $\zeta_{2}$, respectively, and the ζ value at any point C can be expressed as follows:(8)$$\zeta =r\zeta_{1}+\left(1-r\right)\zeta_{2}$$
where the fatigue functional exponents represent the material fracture properties of the model. Note that $\zeta_{2}=$ 1.0 can be fixed to meet Equation (5) without loss of generality; the tensile yield line is not violated. Meanwhile, $\zeta_{1}$ can be calculated by minimizing the fitting error. For the Goodman–Haigh and Gerber diagrams, $\zeta_{1}=\zeta_{2}=$ 1.0 and $\zeta_{1}=\zeta_{2}=$ 2.0, respectively.

Based on observation shown on the left side of Figure 11, we extended the equivalent fatigue strength curve denoted by $\overset{⏜}{\mathrm{EAF}}$ by adding a line connecting points D and E. This represents the linearization of an extended Gerber diagram, achieved using the following:(9)$$\frac{\sigma_{a}}{\sigma_{FS}}-{\left(\frac{\sigma_{m}}{\sigma_{YC}}\right)}^{2}=1$$
and defined by parameter $F_{c1}$. Note that most researchers use unity for this parameter.

As for the left side, we defined the $\sigma_{m}$-$\sigma_{a}$ line for the negative mean stress as a linear combination of the two lines, i.e., the equivalent fatigue strength line $\bar{\mathrm{DE}}$ and the line denoting the tensile yield line $\bar{\mathrm{GH}}$. The directional vector (**d** in Figure 11) of the equivalent line for a stress combination at the point $C^{\prime }$ was interpolated from two fixed directions denoted by the unit vectors $u_{\mathrm{DE}}$ and $u_{\mathrm{GH}}$, using the normalized distance parameter *r*, as follows:(10)$$d=\frac{ru_{\mathrm{GH}}+\left(1-r\right)u_{\mathrm{DE}}}{\left|ru_{\mathrm{GH}}+\left(1-r\right)u_{\mathrm{DE}}\right|}$$

This procedure is conceptually similar to that applied when the mean stress is positive. The line denoting the tensile yield line may be replaced by another approximation to appropriately modulate the directional vector of the $\sigma_{m}$-$\sigma_{a}$ line. Notably, the upper side of the equivalent fatigue strength curve defined by points $\overset{⏜}{\mathrm{DEF}}$ is key in terms of die design for metal forming. In the case of structural design, the region below the curve is more important.

## 4. Validation and Application

Tanrikulu and Karakuzu [2] experimentally explored the HCF fracture behavior of WC-Co20% of a die material during cold forging using a three-point bending test (*R* = 0.1). They predicted the fatigue life of a die used for automatic multi-stage cold forging of a simple flange bolt, using the traditional Goodman–Haigh diagram. Moreover, they used seven combinations of $\sigma_{m}$ and $\sigma_{a}$ and calculated their corresponding life cycles denoted by *N* and equivalent stress amplitudes on the $\sigma_{a}$-axis, denoted as ${\bar{\sigma }}_{a}$, as summarized in Table 2 and the fifth column of Table 3. Using this fatigue life criterion, the predicted die life was 136,700 cycles for $\sigma_{m}$ = 84 MPa and $\sigma_{a}$ = 1038 MPa.

In accordance with Tanrikulu and Karakuzu, we used a transverse rupture stress of 2900 MPa, rather than the tensile strength, when comparing our new approach with the traditional one. We also used their compressive yield strength (3500 MPa). First, after setting $\zeta_{1}=\zeta_{2}=$ 1.0 and $F_{c1}=$ 1.0 (the Goodman–Haigh diagram), we compared our predictions to those of Tanrikulu and Karakuzu. They calculated life cycles using the Basquin law [23].

The predicted fatigue life *N* in this study was calculated using the following formula of assumed linearity (See Figure 12) between equivalent stress amplitude ${\bar{\sigma }}_{a}$ and logarithmic fatigue life log(N) as follows:(11)$${\bar{\sigma }}_{a}=-K\;\log(N)+\sigma_{0}$$
where *K* and $\sigma_{0}$ are calculated from two sets of the experimental equivalent stress amplitudes and logarithmic fatigue lives.

The small differences reflect variations in the estimated equivalent stress amplitudes ${\bar{\sigma }}_{a}$’s and fatigue strength (Table 2 and Table 4) attributable to differences in the image-processing techniques used to acquire the values of $\sigma_{m}$ and $\sigma_{a}$ [2].

We calculated ${\bar{\sigma }}_{a}$’s for the Gerber diagram ($\zeta_{1}=\zeta_{2}=$ 2) and $F_{c1}=$ 2.65 (See Figure 12) for all combinations of $\sigma_{a}$ and $\sigma_{m}$ (see sixth column of Table 3). They are clearly smaller than those of the Goodman–Haigh diagram (fifth column of Table 3). The differences increase as $\sigma_{m}$ or $\sigma_{a}$ increases. The difference at $\sigma_{a}$ = 1830 MPa is about 35% relative to ${\bar{\sigma }}_{a}$ (2830 MPa) of the Goodman–Haigh model. Applying the equivalent stress amplitudes to the cycles of the Gerber diagram, a life cycle of 26,052 episodes was predicted for the stress combination used by Tanrikulu and Karakuzu. This is shown in Table 4 and compared with the values derived using other methods. The value differs greatly from that derived using the Goodman–Haigh diagram (136,301), because the latter diagram underestimates the equivalent stress amplitudes except when *R* = −1.0. The use of an appropriate fatigue curve or line is essential when predicting the HCF life of cold forging dies.

Next, we used a combined Goodman–Haigh and Gerber diagram to solve the same problem. We used the following values: $\zeta_{1}=$ 2.0 and $\zeta_{2}=$ 1.0, $\zeta_{1}=$ 1.5 and $\zeta_{2}=$ 1.0, and $\zeta_{1}=$ 1.25 and $\zeta_{2}=$ 1.0. The equivalent stress amplitudes are listed in columns 7–9 of Table 3, and the respective *r*-values in columns 10–12. Table 3 shows that the equivalent fatigue line of our diagram at the low $\sigma_{a}$ becomes more similar to that of the Gerber diagram (compared to the Goodman–Haigh diagram) as $\zeta_{1}$ increases from 1.0 to 2.0, and that the *r*-value is a function of the $\sigma_{m}$ or $\sigma_{a}$. For all three cases, the predicted life cycles are summarized in Table 4, revealing that these increased from 26,052 to 136,301 as the exponent $\zeta_{1}$ decreased (i.e., from the Gerber to the Goodman–Haigh value). This is explained by the fact that the trial stress case included a small mean stress ($\sigma_{a}$ = 84 MPa), whereas the mean stresses of the experimental points used to evaluate HCF behavior were large (because *R* = 0.1). This emphasizes the critical influence of the die material exponent $\zeta_{1}$, which must be derived using an appropriate method.

We used the data in the fifth and eighth columns of Table 3 to predict the fatigue life of the die marked by S in the nut-forming process shown in Figure 1. The signed effective stress was used to calculate the basis stress. For this purpose, we conducted a HCF test of WC-Co20% under the conditions of $\sigma_{m\;}$ = 0 and $\sigma_{m\;}$ = −500 MPa, as shown in Figure 12. The experiments show that the HCF test specimens fractured after one million cycles both at $\sigma_{m}$ = 0 and $\sigma_{a}$ = 900 MPa (*R* = −1) and at $\sigma_{m}=$ −500 MPa and $\sigma_{a}=$ 1200 MPa (*R* = −2.44). We assessed the effect of a negative mean stress on the equivalent fatigue strength on the $\sigma_{a}$-axis, indicating that a 500 MPa decrease in the mean stress causes a 300 MPa increase in the equivalent fatigue strength.

We thus assumed that the slope of line DE in Figure 11 was −300/500, equivalent to an $F_{c1}=$ 2.65. This is greater than expected, because the mean stress affects brittle material more than ductile material [27]. Figure 13 shows the Goodman–Haigh diagram, and our diagram, for $\zeta_{1}=$ 1.5 and $\zeta_{2}=$ 1.0 and $F_{c1}=$2.65.

We applied the presented MSCD diagram and the Goodman–Haigh diagram to predict qualitatively the HCF life span of the die insert marked by S in Figure 5 in automatic multi-stage cold forging of an automotive nut. The die geometry has 6 planes of symmetry, as shown in Figure 14a. However, because of the features of short billet mechanically sheared shown in Figure 14b, the entire process has no symmetric plane. This non-symmetry caused the HCF fracture at a certain part, as shown in Figure 14a, because the change of billet orientation during the process including shearing stage was negligible. However, we assumed that there were 6 symmetric planes for the accurate prediction of die stresses for the forging simulation but that the traction vectors exerted on the die surface, calculated from the forging simulation, were multiplied by 1.2 to reflect qualitatively the non-symmetry of the process during die structural analysis.

Figure 15 compares the HCF life predicted by the modified Goodman–Haigh diagram and our MSCD or fatigue limit diagram for the same sample. The predicted minimum HCF lives were 45,000 and 100,000 cycles, respectively. The former prediction is more conservative, even though its equivalent fatigue strength (780 MPa) is much greater than that of the latter (670 MPa), because *R* = 0.1 [2]. This is the effect of the slope (−300 MPa/500 MPa) applied to the latter model only.

Compared to the former model, the potential HCF fracture region of the latter model was clearly concentrated in a limited region. This effect, and the HCF life, were qualitatively similar to the experimental data (cf. Figure 15b,c). This reflects the effect of the slope of the equivalent fatigue line in the second quadrant, emphasizing the importance of the mean stress in the HCF fatigue behavior in that quadrant. It is possible to predict the actual HCF life of dies used for metal forming.

## 5. Conclusions

We investigated the characteristics of die stresses developing during cold forging, with an emphasis on the shrink-fitting used to prestress dies prior to cold forging. We found that all the $\sigma_{m}$-$\sigma_{a}$ combinations of a die part exhibited near-identical oscillation patterns, moving toward the $\sigma_{a}$-axis in the MSCD from the prestressed to the final state. The stresses of $\sigma_{m}$ and $\sigma_{a}$ on appropriately prestressed die inserts remained in the second quadrant, which is not the region of interest of structural engineers. On the contrary, stresses on the shrink rings remained in the first quadrant. The stresses of non-prestressed die parts (including punches and ejectors) moved from its initial state (at the origin of the MSCD) toward the line denoting the compressive yield line in the second quadrant.

It was believed that the use of a traditional MSCD is inappropriate for cold forging, based not only on our new data on die mechanics, but also on experimental literature data. To the contrary, our MSCD is general and flexible. It includes the Goodman–Haigh and Gerber diagrams as special cases. We used two material fatigue parameters for the MSCD in the first quadrant. Both were exponents of the mean stress per unit tensile strength. One refers to the fatigue strength curve or line, and the other to the line for the tensile yield line.

A straight line was used to extend the MSCD of the first quadrant to the second quadrant, based on experimental literature data. Its slope of the example, which is a fatigue property, was relatively steep, indicating that the zero-slope assumption of the modified Goodman–Haigh diagram may lead to considerable error when calculating HCF life, especially that of a prestressed die used for cold forging.

Our fatigue limit diagram was numerically tested under various conditions, and its validity and flexibility were verified. The tests revealed the importance of the second MSCD quadrant when predicting the HCF life of prestressed dies which has been neglected by structural engineers. The tests emphasized that fatigue tests on die materials should include at least the negative mean stress case. This determines the slope of the equivalent fatigue line, which is most important when predicting the HCF life of prestressed dies used during metal forming.

## Figures and Tables

**Figure 1 materials-15-02351-f001:**
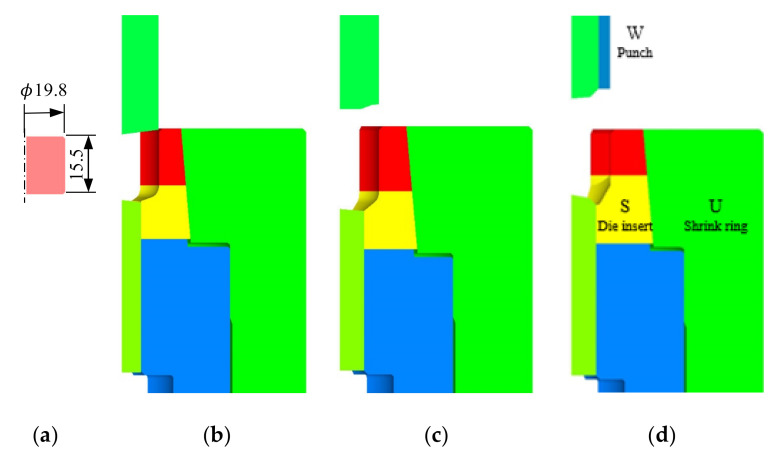
Initial workpiece and die designs of the selected nut cold forging stages: (**a**) workpiece; (**b**) stage 1; (**c**) stage 2; (**d**) stage 3.

**Figure 2 materials-15-02351-f002:**
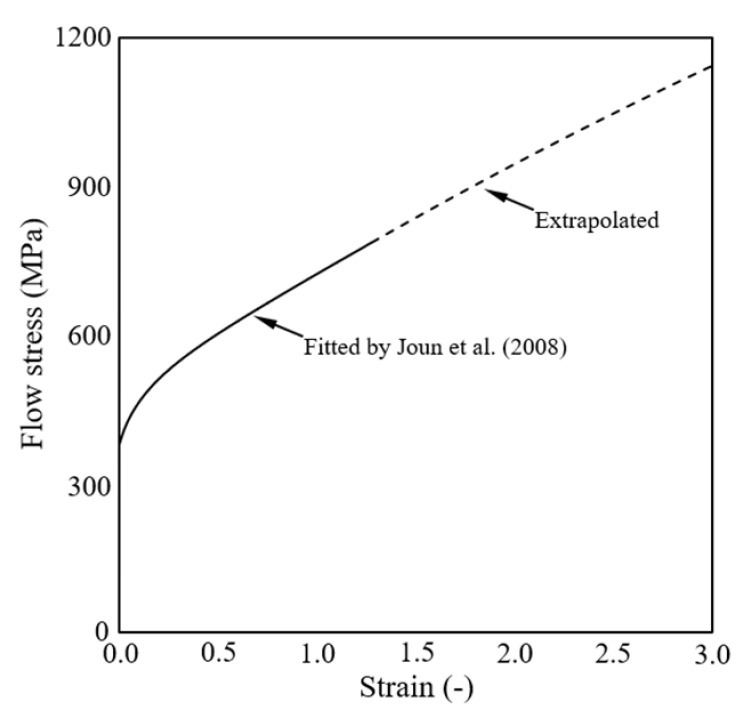
Flow curve of the material S25C.

**Figure 3 materials-15-02351-f003:**
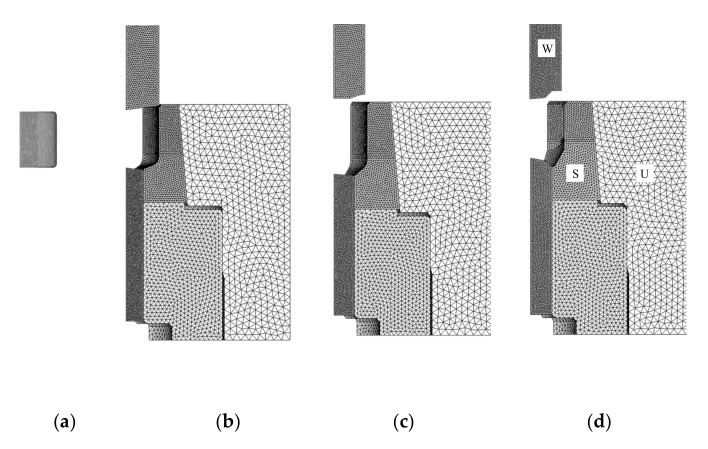
Typical finite element mesh systems of the material and the selected die parts marked by letters S, U and W in Figure 1: (**a**) workpiece; (**b**) stage 1; (**c**) stage 2; (**d**) stage 3.

**Figure 4 materials-15-02351-f004:**
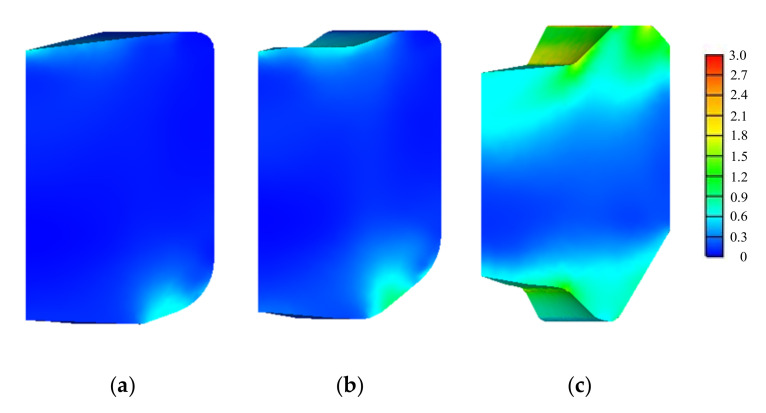
Finite element predictions (Effective strain): (**a**) stage 1; (**b**) stage 2; (**c**) stage 3.

**Figure 5 materials-15-02351-f005:**
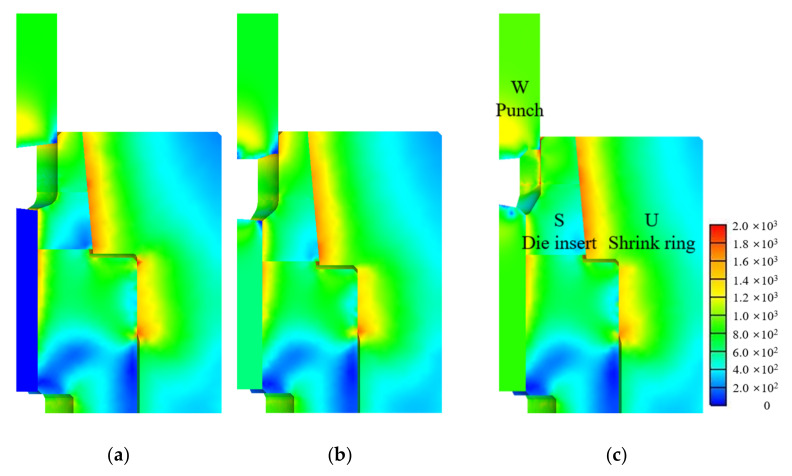
Predicted effective stresses on the die parts at the end of each stage: (**a**) stage 1; (**b**) stage 2; (**c**) stage 3.

**Figure 6 materials-15-02351-f006:**
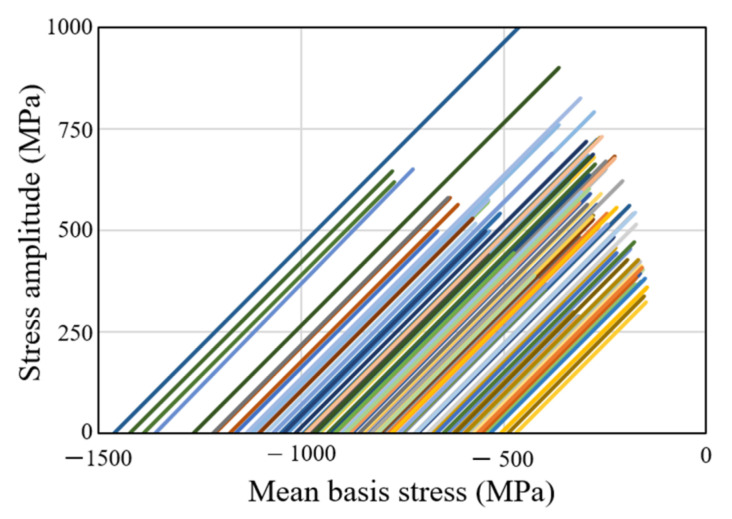
Cyclic stress features of the prestressed die insert S.

**Figure 7 materials-15-02351-f007:**
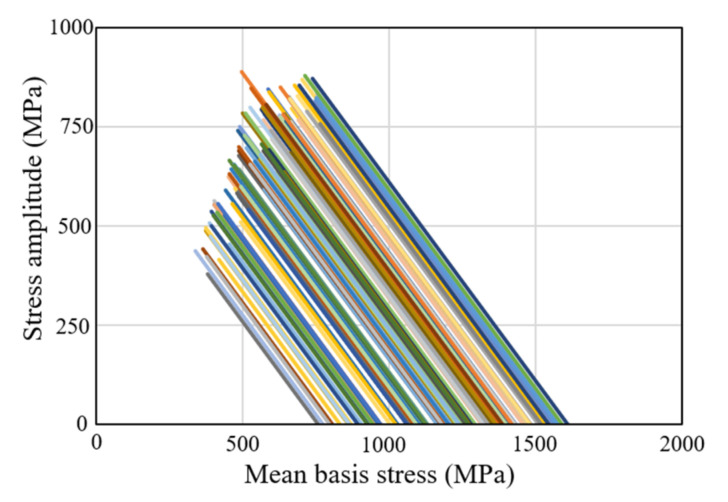
Cyclic stress features of shrink ring U.

**Figure 8 materials-15-02351-f008:**
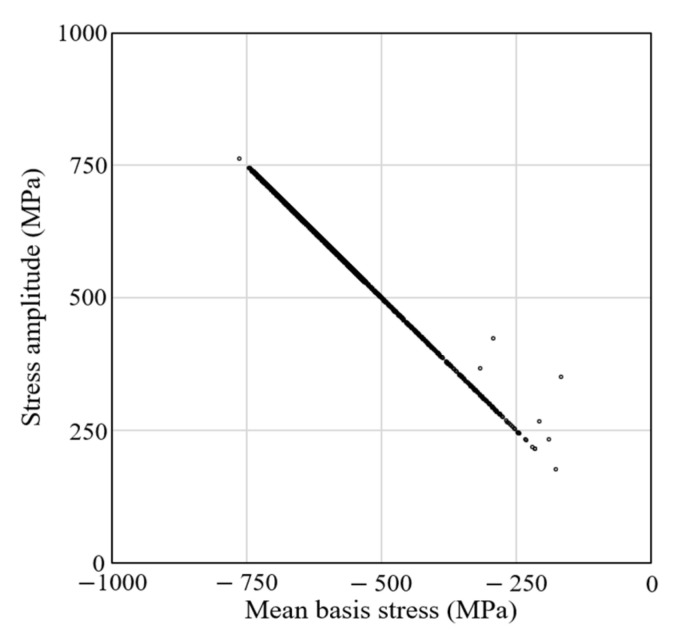
Cyclic stress features of the non-prestressed die parts W.

**Figure 9 materials-15-02351-f009:**
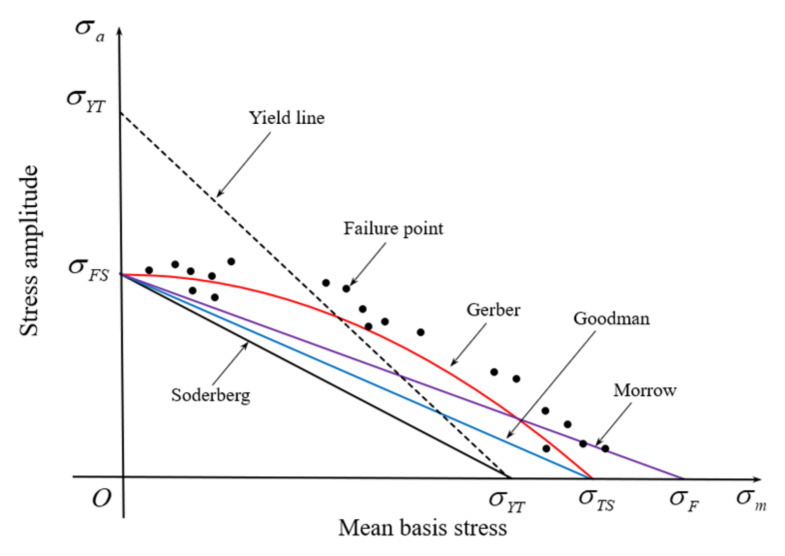
Effect of the mean stress on the equivalent fatigue strength curve [27].

**Figure 10 materials-15-02351-f010:**
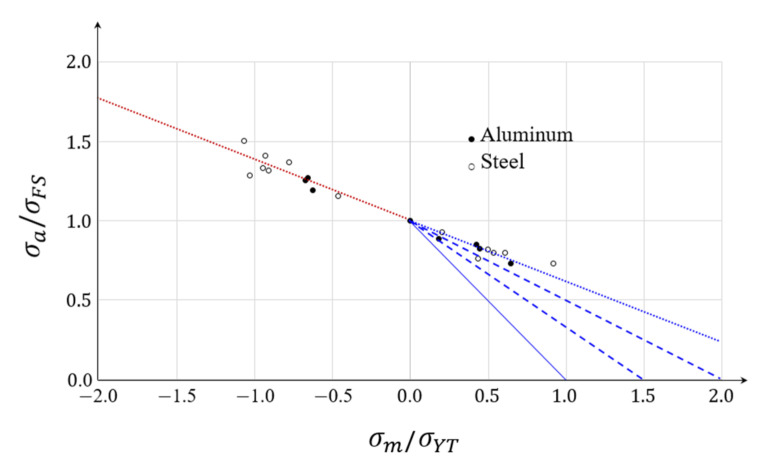
Effects of compressive and tensile mean stresses [6,34]. (●) Aluminum alloys (○) Steels.

**Figure 11 materials-15-02351-f011:**
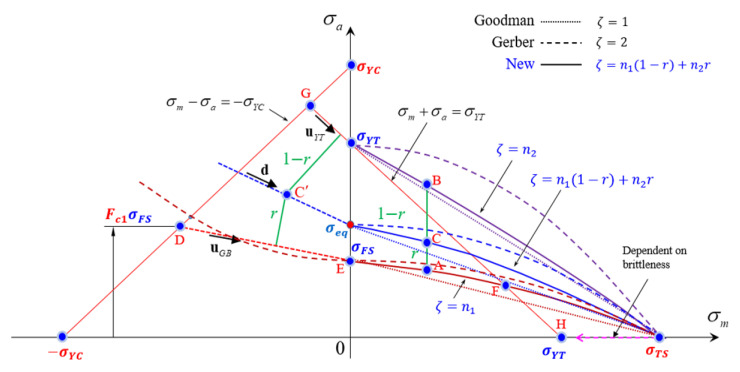
A generalized fatigue limit diagram.

**Figure 12 materials-15-02351-f012:**
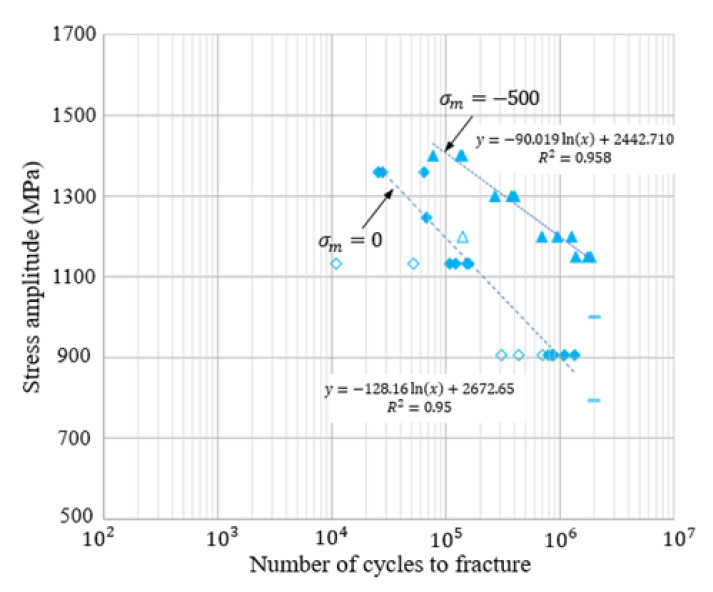
HCF test under zero and negative mean stresses of WC-Co20%. No-fill triangles and diamonds were discarded because of their less reliability in fitting the experiments.

**Figure 13 materials-15-02351-f013:**
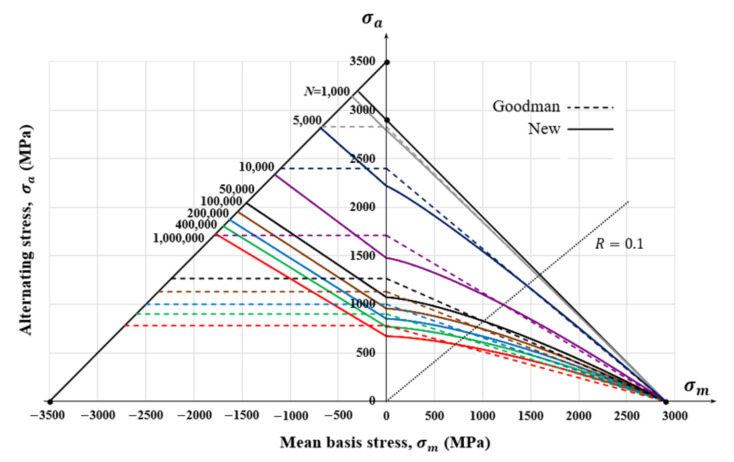
Goodman–Haigh diagram (dashed lines) and the presented MSCD (solid lines for the case of $\zeta_{1}=$ 1.5, $\zeta_{2}=$ 1.0 and $F_{c1}=$ 2.65).

**Figure 14 materials-15-02351-f014:**
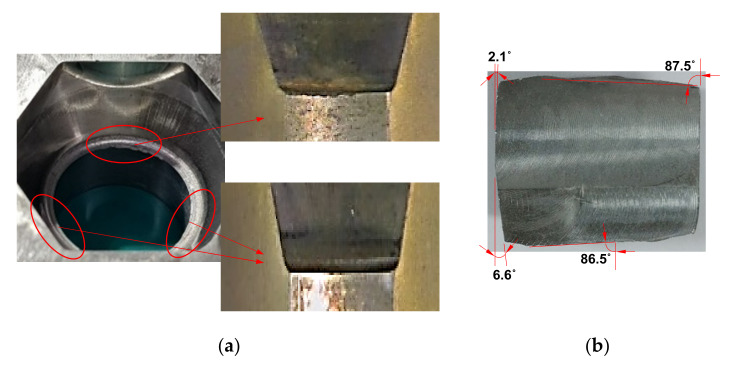
Non-symmetry of the process in terms of HCF fracture: (**a**) Unsymmetric HCF fracture of die insert; (**b**) Initial material.

**Figure 15 materials-15-02351-f015:**
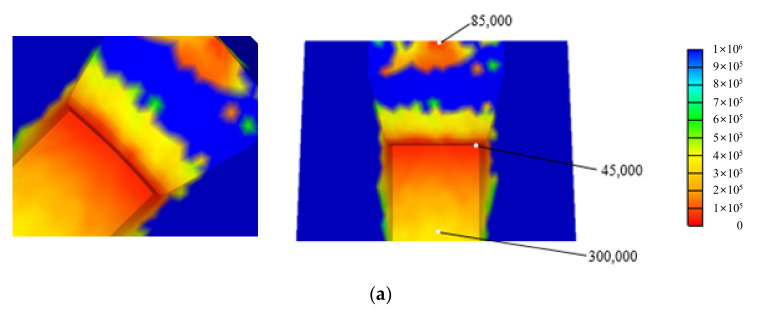
Predicted HCF life of the die insert (marked by S) involved in nut-forming: (**a**) Goodman–Haigh model; (**b**) Our model; (**c**) Fractured die.

**Table 1 materials-15-02351-t001:** Mechanical properties of the die materials.

Material	Young’s Modulus	Poisson Ratio
WC-Co20% [12]	480 GPa	0.26
SHK51 [8]	219 GPa	0.28

**Table 2 materials-15-02351-t002:** Experimental data of the polished surface case of Tanrikulu and Karakuzu [2].

*R*	$\sigma_{a}\;(\mathrm{MPa})$	$\sigma_{m}\;(\mathrm{MPa})$	*N*	${\bar{\sigma }}_{a}\;(\mathrm{MPa})$
0.1	587	716	1,000,000	780
0.1	653	795	400,000	900
0.1	705	857	200,000	1001
0.1	764	934	100,000	1127
0.1	825	1009	50,000*	1265
0.1	994	1217	10,000	1712
0.1	1291	1576	1000	2828

* *N* = 50,000 was calculated by interpolation.

**Table 3 materials-15-02351-t003:** The ${\bar{\sigma }}_{a}$ values obtained using various combinations of $\sigma_{m}$ and $\sigma_{a}$.

*R*	$\sigma_{a}$ (Mpa)	$\sigma_{m}$ (Mpa)	*N*	${\bar{\sigma }}_{a}$ (Mpa)	${\bar{\sigma }}_{a}$ (Mpa)	${\bar{\sigma }}_{a}\;\left(\zeta_{2}=1.0\right)$			*r*-Values $\left(\zeta_{2}=1.0\right)$		
	[2]	[2]	[2]	Goodman	Gerber	$\zeta_{1}=2.0$	$\zeta_{1}=1.5$	$\zeta_{1}=1.25$	$\zeta_{1}=2.0$	$\zeta_{1}=1.5$	$\zeta_{1}=1.25$
0.1	587.4	716.1	Million	780.0	625.5	625.5	669.6	711.2	0.0	0.0	0.0
0.1	652.9	795.0	400 k	899.5	706.0	709.7	766.7	818.0	0.05	0.05	0.05
0.1	705.2	856.8	200 k	1000.9	772.6	781.4	850.0	909.4	0.09	0.10	0.10
0.1	764.3	933.8	100 k	1127.3	852.7	870.7	954.5	1024.1	0.14	0.15	0.16
0.1	824.9	1009.2	50 k	1265.2	938.6	970.9	1071.3	1151.1	0.21	0.22	0.22
0.1	993.5	1217.0	10 k	1712.0	1205.9	1327.0	1475.6	1579.3	0.41	0.42	0.43
0.1	1290.8	1576.4	1 k	2828.1	1832.2	2754.7	2792.3	2810.5	0.96	0.96	0.97

**Table 4 materials-15-02351-t004:** Life cycle predictions of the various MSCDs.

Method	Life Cycles	Fatigue Strength
Goodman model, as modified by Tanrikulu and Karakuzu	136,700	1070
Goodman model used in this paper, $\zeta_{1}=1.0$ and $\zeta_{2}=1.0$	136,300	1069
Gerber model used in this paper, $\zeta_{1}=2.0$ and $\zeta_{2}=2.0$	26,050	1039
Present model, $\zeta_{1}=2.0$ and $\zeta_{2}=1.0$	41,780	1039
Present model, $\zeta_{1}=1.5$ and $\zeta_{2}=1.0$	83,960	1045
Present model, $\zeta_{1}=1.25$ and $\zeta_{2}=1.0$	102,600	1052

## Data Availability

Not applicable.

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
