# Peer review of "A New General Fatigue Limit Diagram and Its Application of Predicting Die Fatigue Life during Cold Forging"

_materials, 2022, doi:10.3390/ma15072351_

Round 1

Reviewer 1 Report

The manuscript treats an interesting topic in the field of fatigue behavior of metals. It proposes a novel approach to determine the high cycle fatigue lifetime. Experimental and numerical studies were carried our and the results were validated with an existing study from the literature. The manuscript can be thought positively for the publication after considering the remarks below.

Line 30 > Long form of HCF

The introduction gives too much detail on the different concerns of fatigue phenomenon rather than the examined subject. Therefore, it disturbs the flow of the text and the reader may have got lost due to the too much details given. The text should be transformed to a leaner form and the story behind the manuscript should be more highlighted.

Equation 1 friction coefficient μ should be given in the text with the symbol.

Line 138, the definition of the flow stress can be given briefly.

Line 139, more detail on the employed elastoplastic model should be given.

Author Response

We prepared the note of answers for each reviewer.

Thank all the reviewers for valuable comments and recommendations.

Reviewer 2 Report

The present paper deals with modelling of stresses and fatigue life in dies / die inserts used for cold forging, especially considerting the shrink-fitting used to prestress dies before cold-forging. On the example of automotive nuts, results obtained using an own model are presented and exhaustively compared to the state of the art. The authors represent a fatigue limit diagram and verify its validity. Scientific approach, graphical presentation and the conclusions drawn are excellent. Lanugage and grammar are good. I have no significant remarks. 

Author Response

We prepared a file of answers attached.

Thank the reviewer for valuable comments and recommendations.

Reviewer 3 Report

Dear Editor and Authors,

In review, I received a manuscript entitled »A new general fatigue limit diagram and its application of predicting die fatigue life during cold forging” considered for publication in MDPI journal “Materials”.

In the manuscript, authors propose a novel mean stress correction model and diagram to predict the high-cycle fatigue lives of cold forging dies, which focuses on the upper part of the equivalent fatigue strength curve. The authors verify the model on the published data, as well applied to practical cold-forging.

Here I list few comments that can further improve the manuscript, although I feel it is already ready to be published:

- Please explain the abbreviations when used for the first time (e.g. line 30: … high cycle fatigue (HCF)… similar as for LCF.

- The introduction is comprehensive and well written. I wish I would receive more manuscripts like this one. I would maybe suggest a small rewrite: currently, the “WC-Co cemented carbide, a composite of WC and Co, has been used to construct die inserts because of its high strength, although this is at the expense of ductility” is written in the past tense, implying that there is a new material used for this purpose. I suggest converting to simple “WC-Co cemented carbide … is used for die inserts because …”. This is from the fact that the Introduction section then focuses on the WC-Co  

- Line 132: please include a bit of description for an out-of-field audience: “… carbon steel S25C nut …”  

- Line 427: I would suggest moving the inset in Fig. 14(a) on the left side of the main image, and making them bigger. The focus is on the two small adjacent figures, which are now too small to show they are the main focus. Sorry for such small comments, consider this only as a suggestion. There is enough space here to use it.

- Line 446: Conclusions. I find the conclusions section well written and very informative. Thank you.

Author Response

(The authors gave the same response as above.)
